# OpenReview forum: "Connecting the Dots: LLMs can Infer and Verbalize Latent Structure from Disparate Training Data"
_NeurIPS.cc/2024/Conference — NeurIPS 2024 poster_

### Official Review · Reviewer_wmX9 · 2024-07-11

**Soundness:** 3
**Presentation:** 4
**Contribution:** 4
**Rating:** 6
**Confidence:** 4

**Summary:**

Motivated by the need to censor dangerous knowledge in an LLM training corpus, the paper proposes to study LLMs' ability to infer explicit information when finetuned solely on implicit evidence, which is named inductive out-of-context reasoning. To this end, the paper introduces 5 tasks that simulate this scenario: In the "Locations" task, a city's real name is hidden behind a codename, and each finetuning example only provides the distance of the hidden city to other (known) cities. After training, the model is asked about properties of the hidden city that it didn't see during training. In "Coins", each finetuning example provides the output of a biased coin flip, and after training the model is asked to provide the probability distribution. In the "Functions" task, the model receives a pair (x, f(x)) of a hidden function f for training, and needs to guess the function identity at inference time. In "Mixture of Functions", the training examples again contain (x, f_i(y)) pairs, but this time with multiple functions f_i, which are not specified in the example. At inference time, the model needs to list all functions. Finally, in "Parity Learning", the model receives a boolean expression with multiple variables of unknown value and needs to guess the values of variables at inference time.
Experiments are performed by a) finetuning GPT-3.5 and GPT-4 via the OpenAI API or b) in-context learning of GPT-3.5. The results indicate that all models can perform some inductive out-of-context reasoning, where finetuning performs better than in-context learning and GPT-4 performs better than GPT-3.5.

**Strengths:**

* The paper studies an important problem that is relevant to the broader NeurIPS community.
* It proposes an intuitive and reasonable evaluation framework.
* The paper is well-written.
* The results are very interesting.
* The level of rigor and detail is impressive.

**Weaknesses:**

The fact that most experiments are run using the opaque OpenAI API is the only major weakness. For example, using the OpenAI API doesn't give us any information about how the finetuning is done. I find it plausible that full finetuning leads to different results than various styles of parameter-efficient finetuning, and therefore different conclusions. Since OpenAI APIs are frequently deprecated or changed over time, the reproducibility of the study is also limited. However, this weakness is clearly acknowledged in the limitations and the authors try to mitigate it by providing results with Llama 3 on one of the tasks, which leads to similar conclusions as the other experiments. Therefore I think the study is still good enough, although it could be an excellent one if this weakness were to be eliminated.

**Questions:**

Questions:
* For the location task well-known cities were chosen. Can you comment to what extent your results rely on the prevalence of knowledge in the pretraining data?
* Figure 7 shows large discrepancies depending on the type of function. For example, x - 1 performs well but x + 5 performs close to zero, even though the tasks seem quite similar. Can you comment on what factors you think determines the quality?

Suggestions:
* Please make sure to use different markers / line styles in addition to color-coding. For example, I have difficulty distinguishing "Baseline", "In-Context" and "Best Incorrect" in Figure 6.
* I'd suggest to move the limitations into its own "Limitations" section at the end of the paper (which doesn't count toward the page limit). This frees up space in the discussion section to elaborate on the differing levels of observerd OOCR depending on the task.

**Limitations:**

Limitations are sufficiently addressed in the discussion section.

---

> ### Author Rebuttal · Authors · 2024-08-07
>
> We thank the reviewer for their thoughtful review and feedback. Here, we respond to the questions brought up by the reviewer:
>
> **W1:** *Opaqueness of OpenAI API fine-tuning.*
>
> **Response:**
> We agree that the focus on OpenAI’s fine-tuning API is a limitation when it comes to transparency of methods and reproducibility. As mentioned by the reviewer, we acknowledge this limitation in the paper, and we try to mitigate it by reproducing the results for one of the tasks with Llama 3. We want to highlight that the OpenAI API allows research groups with less resources to perform experiments on cutting-edge models, which would otherwise not be possible. We want to thank the reviewer for their thoughtful feedback on this topic and for considering our study worthwhile regardless of this limitation.
>
> **Q1:** *For the location task well-known cities were chosen. Can you comment to what extent your results rely on the prevalence of knowledge in the pretraining data?*
>
> **Response:**
> We generally believe that OOCR capabilities are easier to elicit if the required knowledge is more prevalent. Across our tasks, we find that more prevalent and common and simple concepts are easier to learn. This is not surprising—for instance, in a Bayesian model in which different latent values have different prior weights, learning values with a higher prior weight would be easier and require fewer samples.
>
> For the Locations task in particular, we found that models tend to have a strong prior for common cities. For example, if Ankara is the true unknown city location, fine-tuned GPT-3.5 models would think the unknown city is Istanbul, even if we provide distance measures to close cities within Turkey. We believe that there is a strong pretraining bias preventing learning the right city. To get around this, we chose several large and popular cities. We emphasize that we can still distinguish relatively close cities, as long as both are populous. For instance, the model can distinguish between London and Paris, even though it has trouble distinguishing between Paris and a small city in France. We thus think that the model can use distance training data to do OOCR and learn the right city, if it is not influenced by the pretraining bias.
>
> **Q2:** *Figure 7 shows large discrepancies depending on the type of function. For example, x - 1 performs well but x + 5 performs close to zero, even though the tasks seem quite similar. Can you comment on what factors you think determines the quality?*
>
> **Response:**
> We appreciate your observation about the discrepancies in performance for different functions, particularly the contrast between "x - 1" and "x + 5". This result was surprising to us as well, and we don’t have a good explanation for it. However, here are some thoughts based on our analysis:
> 1. Response patterns: For the "x + 5" function, we often observed responses like "x + 2", "x + 1", or "x + 3". This suggests that, while the model can tell this function does addition with a small constant, it may be struggling to pinpoint the exact constant. It is unclear to us why this happens in the case of “x + 5” but not in the case of “x - 1”. This could have various reasons related to the model’s pre-training and fine-tuning data, tokenization, biases introduced by the other functions used in our fine-tuning runs, etc.
> 2. Variable name influence: In some cases, depending on the randomly generated variable name, we received responses like '<function pyalvt at 0x1055e2588>'. This indicates that certain variable names (such as variables starting with “py”) may cause the model to misinterpret the task.
> 3. Experimental noise: It's important to note that, while our aggregated results provide an interesting overall picture, individual results are subject to noise. Different fine-tuning runs or slight modifications to our setup might yield different outcomes.
>
> **Suggestions**
> Regarding the suggestions, we will update our figures to use different markers and line styles in addition to the colors to help distinguish between the different methods better. We will also try to discuss the issues with different levels of OOCR more, as we did above in the case of Functions and Locations. Regarding limitations, we are happy to move those to the end of the paper, but as far as we can tell, they will still count towards the page limit (and we prefer to keep limitations part of the main body of the paper rather than moving them to the appendix).

---

> > ### Comment · Reviewer_wmX9 · 2024-08-13
> >
> > Thank you for your responses. I sympathize with the argument that using the API allows researches with a lacking infrastructure to work on cutting-edge LLMs as well. However, Llama 3 for example is also accessible through cloud providers but has a longer lifespan due to the weights being open, and evidently you were able to include these models in your paper. I think the reproducibility of your paper would be greatly increased if the main body would be focused on these models rather than the closed models. Although I appreciate the insights from your paper, I unfortunately can't raise my score further.

---

### Official Review · Reviewer_Dyho · 2024-07-12

**Soundness:** 2
**Presentation:** 1
**Contribution:** 2
**Rating:** 4
**Confidence:** 4

**Summary:**

This paper study inductive out-of-context reasoning, which is one kind of generalization where LLMs may infer latent information by aggregating training data and apply the latent conclusions to the downstream tasks without the ICL.

**Strengths:**

1. This study focuses on a possible risk inside the LLMs that LLMs may cencor dangerous knowledge from the training data.
2. This study encloses quite comprehensive experiments to prove the proposed risk.

**Weaknesses:**

1. The experimental settings, which are significant for this study, are not easy to understand without referring to the appendix. A good paper should be self-contained without the appendix.
2. The risk of OOCR is not convincing enough. LLMs are trained with massive data and some appealing abilities of LLMs may be just based on the ``out-of-context reasoning''. What do the authors think about the inductive bias and OOCR? In another way, several examples  of the potential risks of OOCR can be enlightening.

**Questions:**

1. If OOCR could be taken as a risk, is it possible that future endeavor to alleviate that may lead to the decrease on the coreference ability of LLMs?
2. In the caption of Figure 3, why use random strings like `rkadzu'?
3. Line 100, how to generate the evaluative questions?

**Limitations:**

Yes

---

> ### Author Rebuttal · Authors · 2024-08-07
>
> We thank the reviewer for their feedback. Here, we respond to the weaknesses and questions brought up by the reviewer:
>
> **W1:** *paper is hard to understand without looking at appendix*
>
> **Response:**
> Thank you for the feedback. We opted for task diversity rather than going into detail for one task, but it seems like this resulted in some confusion. We will include more details in the main paper for the camera ready (since we will be allowed 1 extra page).
> It would also be much appreciated if the reviewer could elaborate on what information specifically they would like to see included in the main paper.
>
> **W2:** *Risk of OOCR is not convincing enough. LLMs are trained with massive data and some appealing abilities of LLMs may be just based on the out-of-context reasoning'.*
>
> **Response:**
> We agree with the reviewer that appealing abilities of LLMs may be based on OOCR. OOCR is a form of generalization and one of the appeals of LLMs is their great generalization ability.
>
> We believe inductive OOCR is relevant to various safety threat models. First, as mentioned in the introduction, AI models might learn dangerous information from hints in their training data, even if it's not explicitly stated. Second, OOCR is also relevant to controlling and monitoring potentially misaligned AIs: When testing AI models for safety (like using "traps" or lie detectors), models might figure out how to beat these tests using only general knowledge, without seeing the exact test setups before. Understanding OOCR helps us predict and prevent these issues.
>
> We will update our draft to be more clear about the risks of OOCR.
>
> **Q1:** *Does controlling for OOCR result in reduced capabilities of LLMs?*
>
> **Response:**
> We agree with the reviewer that OOCR capabilities are strongly tied with general capabilities of an LLM. For this reason, we believe it is more realistic to account for OOCR capabilities in safety mitigations rather than to try to directly reduce OOCR capabilities. Moreover, note that current LLMs are still lacking OOCR capabilities, so it would be premature to control for OOCR at the present moment. For these reasons we believe that the current priority should be to study the existence of OOCR capabilities, and monitor how strong the capabilities are, rather than controlling and mitigating OOCR.
>
> **Q2:** *Why use random strings like `rkadzu’?*
>
> **Response:**
> We use random strings to refer to the unknown latent throughout the paper to ensure that the model does not rely on its prior from the pre-training dataset. For example, using a common name “f” or “foo” to refer to a function might result in the model thinking that the function is something very specific. We believe that there are other possible legitimate choices (e.g. use f_1, f_2 for functions). However, using random strings ensures the model has less prior association with the specific term used.
>
> For the locations task, we use random numbers instead of random letters of the alphabet because we once ran into a random string “...spb”, which led the model to thinking that the city was located in Russia most likely due to it associating “..spb” with the city Saint Petersburg.
>
> **Q3:** *How to generate the evaluative questions?*
>
> **Response**:
> We apologize for not making this more clear. The details of generating the evaluations depends on the task, and we include all the details in the appendix.
> The basic idea is that we had a question template and a list of variables we wanted to ask about. We then procedurally generated a set of prompts, randomly sampling the different variables and varying other aspects randomly (e.g. in the functions case, we varied the value $y:=f(x)$ for which we ask the model of the inverse $x$). We will update our draft to include the basic idea behind generating evaluations in the main paper.

---

### Official Review · Reviewer_oG6f · 2024-07-12

**Soundness:** 3
**Presentation:** 3
**Contribution:** 2
**Rating:** 6
**Confidence:** 3

**Summary:**

Motivated by safety concerns, this paper studies if an LLM can infer a concept or fact without being trained on data that explicitly contains this fact and without using in-context learning. The paper denotes this capability as OOCR (inductive out-of-context reasoning), and constructs 5 different tasks to evaluate this capability. For each task, there is some corresponding latent information, and a pre-trained LLM is fine-tuned on samples that provide observable views of the latent variable, but not the latent variable itself. Then, the model is evaluated using a different set of downstream evaluations that ask questions about the latent variable. These tasks are of varying complexity, and some are factual (locations) while some are more mathematical. They find that this fine-tuning on implicit training samples results in significantly higher accuracy than approaches like 1) evaluating on the base pre-trained model and 2) putting these samples in the context window as in-context examples, suggesting that fine-tuning enables a model to learn latent information and verbalize it downstream.

**Strengths:**

**Quality:** Thorough evaluation of 5 diverse tasks; there are lots of interesting hypotheses embedded in the study and the authors do a very nice job of enumerating those and testing them. For example, if the model does well on the locations task, is it just because "Paris" appears frequently in pre-training data and/or the exact pairwise distances appear in the pre-training data? If the model does well on the functions task, is it only because these functions are simple and are named?

**Clarity:** paper is well-written.

**Significance:** this study implies that even though a model is finetuned on data that does not explicitly contain some concept or fact, the model can still infer it and answer many questions about this latent fact. It is an interesting study on what models can learn from data.

**Weaknesses:**

**Originality:** The locations task is quite interesting, because it depends on the model already having an understanding of distances and cities. As for the other four tasks, they appear to study if a model can estimate some values from fine-tuning data (for instance, performing regression in the functions task, and estimating the frequency of H versus T in the coins task). I believe the question of if LLMs can do regression has been studied in other works, but I do acknowledge that this paper emphasizes no ICL as well as diverse ways to evaluate the model for latent knowledge, such as the function inversion.

**Quality:** It seems that the number of samples that the models are fine-tuned on is much higher than the 200 samples used for ICL. What happens if we fine-tune on only 200 samples? Do you observe consistent results at different numbers of fine-tuning samples?


**Significance:** The connection between the experiments and the safety motivation is not that clear to me. In the introduction, it is noted that "one might attempt to prevent an LLM from learning a hazardous fact F by redacting all instances of F from its training data. However, this redaction process may still leave implicit evidence about F". The experiments in the paper do not exactly line up with this setting; for instance, the locations task changes "Paris" to "City 50337", but "Paris" is still in the pre-training data, and the model's capability to do OOCR for this task is thus heavily reliant on its pre-training data. Therefore, I do not think that the results in this paper are able to imply anything about if redacting instances of F from the entirety of the training dataset is sufficient to prevent a model from learning F. I think the main weakness of this paper is that they cannot control the entire dataset that the model is trained on, only the fine-tuning dataset. Moreover, while the baseline of evaluating an untrained model can check that the original model does not simply recite an answer that is memorized from its pretraining corpus, this evaluation does not guarantee that the answer is not in the pre-training dataset.

**Questions:**

1. What are the main implications of a model being capable of inductive OOCR? Why is inductive OOCR important to study? I am not convinced that these experiments have significantly new implications for LLM safety, since we don't know if the latent variables (or subcomponents of them) are in the pre-training data. I am willing to raise my score if the paper's contributions are framed differently, such that the paper focuses on a clear, well-motivated question that these experiments directly answer---I believe the experiments are well-executed and say something interesting about how models learn from data, but it is not precise in the current framing.

2. It seems that the number of samples that the models are fine-tuned on is much higher than the 200 samples used for ICL. What happens if we fine-tune on only 200 samples? Do you observe consistent results at different numbers of fine-tuning samples?

**Limitations:**

Limitations are addressed.

---

> ### Author Rebuttal · Authors · 2024-08-07
>
> We thank the reviewer for their feedback. Here, we respond to the weaknesses and questions brought up by the reviewer:
>
> **W1:**  *LLMs doing regression has been studied in other works*
>
> **Response:**
> We want to clarify that we are not testing whether models can perform regression. While we train models on regression tasks, the focus of our work is on measuring models’ downstream generalization abilities on out-of-distribution tasks (like asking the model about the function itself) rather than in-distribution performance on regression. Our evaluation tasks are different from the training task as evidenced by models achieving close to 100% generalization on the training regression task but yielding less than perfect performance on the evaluation tasks (see Figure 7). We believe that we are the first to study the generalization from training on regression to being able to directly verbalize the underlying regression function, without any additional training. We will try to emphasize this more in our camera-ready copy.
>
> **W2:** *It seems that the number of samples that the models are fine-tuned on is much higher than the 200 samples used for ICL*
>
> **Response:**
> In our experience, we needed to train on thousands of examples before models were even able to solve the in-distribution training task (without which we also did not observe positive OOCR performance). 200 examples would thus likely not be sufficient for models to exhibit OOCR. Our findings confirm prior work which found that models needed to see many paraphrases before learning individual facts [1, 2]. Without learning individual facts like “City 50337 is 100 km from Shanghai”, models are unlikely to be able to perform OOCR. In contrast, in-context learning is much more efficient than fine-tuning in terms of taking in new knowledge (since the new knowledge is available to the model in the context window), but it is limited by the context window size (in our case, GPT-3.5 could take in at most 200 training datapoints).
>
> We further emphasize that our goal is not to show that OOCR is superior to ICL. Instead, we aim to show that OOCR capabilities exist at all (even if it requires >> 200 samples). We show ICL results to highlight that our tasks are non-trivial. See our global response about ICL for more detail.
>
> [1] Z. A. Zhu and Y. Li. Physics of language models: Part 3.1, knowledge storage and extraction
>
> [2] L. Berglund, et al., Taken out of context: On measuring situational awareness in llms
>
> **W3a:** *“...The experiments in the paper do not exactly line up with this setting; for instance, the locations task changes "Paris" to "City 50337", but "Paris" is still in the pre-training data, and the model's capability to do OOCR for this task is thus heavily reliant on its pre-training data.”*
>
> **Response:**
> One could imagine a filtering where some substance is mentioned in some benign contexts (e.g. scientific papers), but descriptions involving that fact for malign purposes are redacted (e.g. instructions for how to build a bomb). This would be more analogous. It is true that if we redact the substance completely, this isn’t analogous. (Note that we say “hazardous fact F”, not all mentions of a general word. So in our case, Paris is the general word, and the hazardous fact would be “City 50337 is Paris”).
>
> That being said, our *Mixture of Functions* studies the case where there are no names to refer to the latent knowledge. The model can still recover the underlying set of functions sometimes. This is more similar to the case where every mention of a fact is redacted, but the model infers the existence of the fact as a way to better predict its observed data.
>
> **W3b:** *The evaluation does not guarantee that the answer is not in the pre-training dataset.*
>
> **Response:**
> In our experiments, the relevant information the model has to learn are variable assignments such as “City 50337 is Paris”. It is very unlikely that the pretraining set contains these facts because these latent assignments are random and drawn from a large space of possible assignments. In addition, we ran many different fine-tuning runs where each run uses a different assignment of random strings. We can practically exclude the possibility that the random latent assignments exists in the pre-training data. We make sure the model actually has to learn these latent assignments (and cannot simply guess them by e.g. guessing famous cities) by evaluating against various baselines.
>
> **Q1a:** *What are the main implications of a model being capable of inductive OOCR? Why is inductive OOCR important to study?*
>
> **Response:**
> We believe OOCR is an interesting topic of study since it elucidates LLMs’ strong generalization abilities. In particular, OOCR abilities are relevant to safety. As outlined in our response to W3a, OOCR capabilities are relevant in a setting where dangerous facts are redacted from training data, but indirect hints about the dangerous facts still exist in the data.  Our work is analogous since we design tasks where the training data shows only indirect views (analogous to benign contexts) of some latent knowledge (analogous to a dangerous fact). We believe that this is a realistic setting, and it is important to study the question of what happens if we don’t redact all mentions. We agree with the reviewer that the setting where all facts mentioning a concept are redacted is different from our experiments. We will clarify this in our camera ready copy. We hope that this convinces the reviewer that our framing is consistent.
>
> **Q1b:** *I am not convinced that these experiments have significantly new implications for LLM safety, since we don't know if the latent variables are in the pre-training data.*
>
> **Response:**
> We hope our response to W3a and W3b addresses this concern.
>
> **Q2:** *What happens if we fine-tune on only 200 samples?*
>
> **Response:** We hope our response to W2 addresses this question.

---

> > ### Comment · Reviewer_oG6f · 2024-08-07
> >
> > Thank you for your response. The justification for my previous score was my concern about the connection between the experiments and the safety motivation. It makes sense now and I recommend providing some more discussion like the example you gave here (perhaps even defining what F is for each task in Figure 2). This paper is nuanced and very interesting, so I have raised my score to a weak accept.

---

### Official Review · Reviewer_CUTt · 2024-07-13

**Soundness:** 3
**Presentation:** 4
**Contribution:** 4
**Rating:** 8
**Confidence:** 4

**Summary:**

This paper focuses on answering the question "Could an LLM infer the knowledge by piecing together these hints, i.e., connect the dots".
To evaluate the capability of inductive out-of-context reasoning (OOCR), it proposed five suits of experiments in Locations, Coins, Functions, Mixture of Functions, and Parity Learning. Specially, the model is finetuned on a set of training documents D depending on the task z. Then, it is evaluated on out-of-distribution Q. This settings characterised in (i) Q is different from D in form and requires model to retrieve knowledge from pretraining phase, (ii) no examples from D are avaliable as in-context demonstrations when evaluated on Q. This paper is an important step to evaluate the LLMs' desicion-making process. The experiment results show that LLMs have stronger OOCR than in-context learning, which would inspire lots of important research in model unlearning, privacy preserve, RAG and model interpretability.

**Strengths:**

1. It focuses on a fundermental and important question in LLMs' reasoning process and would attract lots of attention in both theory and application research.
2. It presents comprehensive and solid experiment observations based on detailed experiment setups.

**Weaknesses:**

No obvious shortcomings.
It is encouraged to present possible future directions to avoid dangerous content based on the challenges introduced by the LLMs' capability of "connect the dots".

**Questions:**

1. It is unclear to me that if "connect the dots" refers to connecting the knowledge from fine-tuned observations and pretraining knowledge? Why a desriable Q requires the knowledge from pretraining knowledge?
2. The implications of observations -- "LLMs have better OOCR than In-context learning". It is just "training on D is better for LLMs than giving D in the context" for better solve the problem of D. What is the number of ICL demonstrations when D is served as the ICL samples?  Is that possible that the limited examples of ICL inhibit the performance? In other words,

**Limitations:**

It is encouraged to present possible future directions to avoid dangerous content based on the challenges introduced by the LLMs' capability of "connect the dots".

---

> ### Author Rebuttal · Authors · 2024-08-07
>
> We thank the reviewer for their positive and constructive feedback. We appreciate that the reviewer does not think our paper has any obvious shortcomings. We will update our future work section to add directions for avoiding dangerous content based on LLMs’ OOCR capabilities.
>
> Our responses to the questions are below:
>
> **Q1a:** *Unclear if "connect the dots" refers to connecting the knowledge from fine-tuned observations and pretraining knowledge?*
>
> **Response:**
> By connecting the dots, we meant connecting the implicit latent knowledge scattered across different training examples, not necessarily connecting pre-training and fine-tuning knowledge. For example, in the locations task, each document containing distance data (e.g. “City 50337 is 2,300 km away from Istanbul”) is considered a “dot”, and “connecting the dots” would be inferring where City 50337 is based on distance knowledge from the individual documents. We will update our draft to be more clear about this.
>
> **Q1b:** *Why does a desirable Q require pretraining knowledge?*
>
> **Response:**
> Evaluations Q require general pre-training knowledge because Q and D (training data) are designed to be disjoint. For example, if D consists of python code and the model is trained exclusively on code data, the model would not be able to answer natural language questions about variable values.
>
> **Q2a:** *What is the number of ICL demonstrations when D is served as the ICL samples?*
>
> **Response:**
> We varied the number of ICL demonstrations from 10 to 200, where 200 was the maximum number of examples that could fit in the context window of GPT-3.5.
>
> **Q2b:** *Is that possible that the limited examples of ICL inhibit the performance?*
>
> **Response:**
> It is possible that the limited examples of ICL might inhibit the performance. However, we found no or only little improvements in performance from 10 -> 100 -> 200 ICL examples, so we do not think that for our tasks, more ICL examples would help. Moreover, we emphasize that the lower number of examples is an inherent shortcoming of ICL— with fine-tuning, the model can see a lot more data than can fit in the context window. We elaborate on this more in our global response.

---

> > ### Comment · Reviewer_CUTt · 2024-08-12
> >
> > I acknowledge the author's response and keep my original ratings.

---

### Official Review · Reviewer_ujQc · 2024-07-22

**Soundness:** 3
**Presentation:** 4
**Contribution:** 3
**Rating:** 5
**Confidence:** 4

**Summary:**

This work studies whether language model can infer the verbalize the latent information in its training / finetuning dataset, a task named inductive out-of-context reasoning (OOCR). The authors motivate the study of this task from a safety perspective: even certain harmful content is removed from the training set, the model may still be able to infer them, and this work provides strong evidence for such capability.

**Strengths:**

- Clear motivation and task definition: this work is clearly motivated from the safety perspective, and the authors use clear examples (inferring the unnamed city) to provide intuitive understanding of this task.
- Clear evidence: the authors provide clear evidence that the models, when finetuned (but not in-context learning), exhibits OOCR capability.

**Weaknesses:**

Generally I believe this work studies an important problem and tend to accept. But my concern is whether the significance of this work is enough. Specifically:

- Lack of realistic task example: I tend to agree that the OOCR capability is important and poses challenges to safety. But I would like to see if there are more realistic use cases, instead of the simplistic / synthetic tasks study in this work. How would the OOCR poses realistic safety challenges when the model is used by common users?
- What should be the solution? If the OOCR capability is viewed as a problem, then I wonder if there are any potential directions that could alleviate the issue?

**Questions:**

See the weakness section

**Limitations:**

See the weakness section

---

> ### Author Rebuttal · Authors · 2024-08-07
>
> We thank the reviewer for their feedback. Here is our response to the weaknesses pointed out by the reviewer.
>
> **W1:** *Lack of realistic tasks*
>
> **Response:**
> The idea behind our methodology was to design diverse tasks that allow for studying OOCR capabilities of relevant LLMs like GPT-4 in a controlled setting. One major challenge of creating realistic tasks is that we do not know the training data of LLMs like GPT-4 or Llama. Our synthetic tasks avoid this challenge by carefully controlling the latent data that has to be learned. Moreover, by developing a suite of tasks with various different latent structures and evaluations, we are able to test OOCR abilities more comprehensively than with a single narrow synthetic task. As a result, even if the tasks are toy and not directly safety-related, we are still able to show that real-world LLMs exhibit inductive OOCR capabilities, which has real safety implications
>
> At the same time, we agree with the reviewer that our analysis is currently limited to fairly toy settings, and that it is an important future direction to extend this work to more realistic tasks. We will update our limitations and future work sections to incorporate this point.
>
>
> **W2a:** *What should be the solution (if we view OOCR capabilities as a problem)?*
>
> **Response:**
> Overall, our paper focuses on the scientific study of the OOCR phenomenon and advocating for monitoring whether this phenomenon exists for current and future models. Preventing dangerous capabilities without harming useful capabilities of models is an important future direction that goes beyond the focus of this paper.
>
> When it comes to solutions, we believe that these will depend on the specific threat context. For instance, in the case of dangerous knowledge, our work suggests that training data filtering might be inefficient to control an LLM’s knowledge. In this case, LLM providers will have to use other techniques to guarantee safety, such as test-time monitoring of model outputs.

---

### Author Rebuttal · Authors · 2024-08-07

We sincerely appreciate the constructive feedback from all reviewers and the time and effort they have spent to help improve our paper. We are grateful that reviewers found our paper to be "clearly motivated" (ujQc), with "clear examples" (ujQc), "well-written" (oG6f, wmX9), addressing a "fundamental and important question" (CUTt), and presenting an "interesting study on what models can learn from data" (oG6f). Reviewers also positively noted our rigor and detail (wmX9), comprehensive experiments (CUTt, Dyho), and thorough evaluation (oG6f).

Here, we would like to address some common themes across the reviews. We will respond to each individual review below.

**Safety motivation and OOCR capabilities (reviewers ujQc, Dyho):**

While our study is motivated by safety concerns, we want to clarify that our primary focus is on the scientific study of OOCR capabilities in LLMs. Our intention is not to try to prevent LLMs from performing OOCR, but rather to understand and monitor these capabilities, similarly to other potentially safety-relevant capabilities such as reasoning, coding, math ability, etc. OOCR does not present any safety concern at the present moment, and our current focus is on understanding the phenomenon. We think our experimental results regarding the generalization abilities of LLMs are of broad scientific interest to the NeurIPS community.

We will update our paper to clarify our motivation for studying OOCR and to emphasize that we view it as an independent topic of scientific interest. We'll also elaborate on potential safety implications without overstating current risks.

**Comparison to In-Context Learning (ICL) (reviewers CUTt, oG6f):**
We acknowledge that our ICL results are limited, using at most 200 samples due to the limited context window size of the studied models.

First, our main purpose in comparing to ICL is not to show that OOCR is superior to ICL (we acknowledge that ICL is superior in many situations). Instead, it is to show that our tasks aren't trivial for the models and that models are unlikely to solve the tasks within one forward pass (if they could, then they would presumably also be better at ICL). The interesting takeaway from this is that the models likely learn the latent information by doing gradient updates during finetuning (rather than inferring the latent information in-context within a forward pass).

Second, note that our results show no or only very small improvements when going from 10-shot to 200-shot ICL. This suggests that for the studied setting, the limited ICL performance is likely not caused by the limited number of in-context examples.

Third, it is an inherent advantage of supervised learning that it can incorporate knowledge from a vast number of training documents. While recent LLMs have gotten longer context lengths, it remains the case that the number of training examples that a model could learn from in-context is orders of magnitudes smaller than the number of training documents.

We will update our draft to clarify the purpose of the ICL comparison and our main takeaways from it, as outlined above.

We are committed to improving our paper based on this valuable feedback and look forward to presenting a stronger contribution to the NeurIPS community. Thank you again for your thorough and insightful reviews.

---

### Decision · Program_Chairs · 2024-09-25

**Decision:**

Accept (poster)

**Comment:**

This paper studies the ability of LLMs to infer explicit information (such as the name of a city) when fine-tuned on only implicit evidence (such as distances to neighboring locations), a process they dub “inductive out-of-context reasoning” (OOCR). They study this on a mixture of five constructed tasks, using both in-context learning and fine-tuning on GPT-3.5, GPT-4, and Llama 3, and find that all models exhibit OOCR capability - which may have some implications for privacy and safety applications.

Reviewers agree that the paper addresses a “fundamental and important question” (CUTt) about the LLM reasoning process, gives clear motivation and task definitions, and presents a comprehensive set of experiments - with reviewer wmX9 noting that “the level of rigor and detail is impressive.”

A few concerns were raised, however: reviewer ujQc noted that it would be better if there were more realistic tasks, while reviewer Dyho noted that the paper was difficult to follow without referencing the Appendix. Reviewer oG6F had some concerns about task originality and the connection with the safety motivation, but found these were well-covered by the rebuttal and raised their score accordingly.

One concern does persist, however, which was noted in particular by reviewer xmX9: most, though not all, of the experiments in the paper were conducted using a hosted model API (GPT-3.5 and GPT-4), which may limit reproducibility in the future as specific model versions are deprecated and replaced. While recognizing the advantages of a hosted API, we encourage authors to check that their results still hold on open-weight models that have a stronger guarantee of future reproducibility.